

# Distinguishing between enamel fluorosis and other enamel defects in permanent teeth of children

Aira Sabokseir[1], Ali Golkari[1] and Aubrey Sheiham[2,†]

[1] Department of Dental Public Health, Shiraz University of Medical Sciences, Shiraz, Iran
[2] Department of Epidemiology and Public Health, University College London, London, United Kingdom
[†] Deceased.

## ABSTRACT

**Background.** The inconsistent prevalence of fluorosis for a given level of fluoride in drinking water suggests developmental defects of enamel (DDEs) other than fluorosis were being misdiagnosed as fluorosis. The imprecise definition and subjective perception of fluorosis indices could result in misdiagnosis of dental fluorosis. This study was conducted to distinguish genuine fluorosis from fluorosis-resembling defects that could have adverse health-related events as a cause using Early Childhood Events Life-grid method (ECEL).

**Methods.** A study was conducted on 400 9-year-old children from areas with high, optimal and low levels of fluoride in the drinking water of Fars province, Iran. Fluorosis cases were diagnosed on the standardized one view photographs of the anterior teeth using Dean's and TF (Thylstrup and Fejerskov) Indices by calibrated dentists. Agreements between examiners were tested. Early childhood health-related data collected retrospectively by ECEL method were matched with the position of enamel defects.

**Results.** Using both Dean and TF indices three out of four dentists diagnosed that 31.3% (115) children had fluorosis, 58.0%, 29.1%, and 10.0% in high (2.12–2.85 ppm), optimal (0.62–1.22 ppm), and low (0.24–0.29 ppm) fluoride areas respectively ($p < 0.001$). After matching health-related events in the 115 (31.3%) of children diagnosed with fluorosis, 31 (8.4%) of children had fluorosis which could be matched with their adverse health-related events. This suggests that what was diagnosed as fluorosis were non-fluoride related DDEs that resemble fluorosis.

**Discussion.** The frequently used measures of fluorosis appear to overscore fluorosis. Use of ECEL method to consider health related events relevant to DDEs could help to differentiate between genuine fluorosis and fluorosis-resembling defects.

Corresponding author
Ali Golkari, aligolkari@yahoo.com, golkaria@sums.ac.ir

# INTRODUCTION

Despite the extensive use of well documented indices of dental fluorosis (*Dean, 1942*; *Fejerskov, 1988*), there is inconsistency in the reports on the prevalence rates of fluorosis for a given level of fluoride in drinking water. Most probably, this inconsistency in the prevalence of fluorosis occurs due to subjective perception of fluorosis by examiners. Therefore, there is a strong possibility that other Developmental Defects of Enamel

(DDE) rather than excess intake of fluoride are being misdiagnosed as fluorosis (*Atar & Körperich, 2010*).

The two widely-used indices of dental fluorosis are Dean's Index (*Dean, 1934*; *Dean, 1942*) and the Thylstrup and Fejerskov Index (TF Index or TFI) (*Thylstrup & Fejerskov, 1978*; *Fejerskov, 1988*). None of them clearly distinguish between defects caused by fluorosis and caused by other factors. The differences between some of the diagnostic categories are uncertain, vague, or insensitive (a clear example is the "questionable" category in Dean's Index). In Dean's Index, each individual is given one score, as a score for the whole mouth, according to the two teeth most affected by fluorosis. This index categorizes each tooth as normal, questionable, very mild, mild, moderate, moderately severe, and severe. The classification is based on color and extent of discolored enamel together with added hypoplasia in case tooth belongs to the last two categories.

*Thylstrup & Fejerskov (1978)* reformulated Dean' Index as the TF Index (TFI). They used both clinical and histological appearance of fluorosis and created a single coded index from 0 (normal) to 9 . The TFI was modified and finalized in 1988 (*Fejerskov, 1988*). From 1998, the scoring of facial surface was recommended for TFI. Teeth should be cleaned and dried before examination. Cleaning and drying of teeth make the appearance of fluorotic change more prominent and also the diagnosis of questionable cases easier. Still the difference between some categories of TFI is not clear.

As the main indices for diagnosing dental fluorosis include definitions that are imprecise, it is not surprising that in some studies the reported prevalence of fluorosis was similar despite the levels of fluoride in the drinking water being similar or different. For example, a study in KwaNdebele (Africa) revealed that the prevalence of fluorosis was similar in residents of areas with considerably different levels of fluoride in the water supply (*Lewis & Chikte, 1995*). A study in Andhra Pradesh (India) of four different areas with different levels of fluoride in drinking water (<0.7, 0.7–1.2, 1.3–4.0, >4 ppm) reported 100% dental fluorosis even in areas with optimum level of fluoride (*Sudhir et al., 2009*). In Hong Kong, where the fluoride concentration of public water supplies was increased from 0.5 ppm to 0.7 ppm, and then 1 ppm, the prevalence of DDE decreased significantly from 92%, to 55%, and then 35% (*Wong et al., 2006*).

DDEs can be localized or generalized. Numerous systemic risk factors cause generalized or "diffuse" DDEs (*Small & Murray, 1978*; *Pindborg, 1982*; *Atar & Körperich, 2010*). The generalized DDEs may be of genetic origin (*Thesleff, 2000*; *Atar & Körperich, 2010*) or caused by malnutrition or diseases that occurred during early childhood (*Pindborg, 1982*; *Atar & Körperich, 2010*). Fluorosis and some other generalized DDEs are similar in appearance (*Small & Murray, 1978*; *Pindborg, 1982*). The inconsistent reports on the prevalence of fluorosis suggest that non-fluoride related DDEs were being misdiagnosed as fluorosis. Using teeth as markers of significant health-related events and considering these risk factors could help to distinguish between genuine fluorosis and non-fluoride related DDEs that resemble fluorosis.

Genuine fluorosis can best be distinguished from other DDEs by relating a DDE to particular life events, such as a significant health related event. Such events can be reasonably recorded using the life-grid method that helps people remember past events

more accurately (*Blane, 1996*). The life-grid method has been widely used and been very successful in obtaining the precise timing of past events in both qualitative and quantitative studies (*Berney & Blane, 1997*; *Holland et al., 2000*; *Bell, 2005*). A specially designed life-grid for early childhood containing developmental milestones and shorter periods of time for the earlier years, is the Early Childhood Events Life-grid method (ECEL) (*Golkari, 2009*).

As there is a need for accurate data of dental fluorosis, the inconsistency of reports on the prevalence of fluorosis suggests that more precise definitions and diagnostic methods are needed for diagnosing dental fluorosis and distinguishing enamel fluorosis from other DDEs. Therefore, a study was planned with the objective of distinguishing genuine fluorosis from fluorosis-resembling defects that could have adverse health-related events as a cause, instead of just excess fluoride intake.

## MATERIALS AND METHODS

This cross-sectional study was performed on 400 9-year-old children of Fars province, Iran. The children were randomly selected from areas of Iran with high, optimal and low levels of fluoride in the drinking water. Fluorosis cases were diagnosed using Dean's Index and TF Index (*Dean, 1942*; *Fejerskov, 1988*) by calibrated dentists. To differentiate between genuine fluorosis and fluorosis resembling defects, early childhood data were collected retrospectively by the ECEL method (*Golkari, 2009*).

Ethical permission was obtained from the Ethical Committee of Shiraz University of Medical Sciences (SUMS) and the Educational Head Office of the Fars Province. A written consent form explaining the objectives and the stages of the study were sent to the parents of the selected children. They were included only if the parents agreed to take part.

The level of fluoride in water of each town of the province was obtained from their primary health care trust. Towns were categorized into one of three fluoride categories; high, optimal, and low fluoride. One town was selected randomly from each category. The selected towns were Gerash with high fluoride (2.12–2.85 ppm) in water, Sepidan with low fluoride (0.24–0.29 ppm) in water, and Shiraz was the city with an optimal range of fluoride (0.62–1.22 ppm) in water.

Sample size was calculated based on the estimated prevalence of fluorosis in Iran (61%), $d = 6.1\%$ (10% expected prevalence), $\alpha = 0.05$. As a result, a sample size of 246 was needed if a simple randomized sampling was used in all areas. The selection of children in Gerash and Sepidan (with high and low fluoride levels) was done by simple randomized sampling. However, in Shiraz (chosen as area with optimum fluoride level), the simple randomization was not possible as it was a big city. Therefore, a stratified randomization method had to be adopted. It was done using the four different educational zones of the city. The number of required sample in this area was multiplied by 2 ($k = 2$) to increase the accuracy in the stratified sampling method that was used. The response rate was suggested to be around 80%. Therefore, at the end, 100 9-year-old schoolchildren of Gerash, 100 from Sepidan and 200 from Shiraz were selected from lists of students obtained from Educational Head Office of each town.

Children aged from nine years, 0 month to nine years and 11 months who returned the signed consent form were included in the study. Children were excluded if they had lived

for more than six months from birth to five years of age in other towns (determined during interview with parents). Those who had less than seven permanent incisor teeth, those with orthodontic brackets, overlapping teeth, and large restorations or severe extrinsic stains on their incisors were also excluded (determined during intra-oral examination).

Intra-oral examinations were carried out to select children conforming to the study criteria. The examinations were performed using a headlight, disposable mirrors and tongue blades with children seated on a chair. Photographs of the dentition of selected children were taken for the diagnosis of dental fluorosis. A one-view photograph was taken of the anterior part of dentition using a digital camera (Nikon D7100, AF-S VR Micro-Nikkor 105 mm f/2.8 G IF-ED) based on methods described by *Wong et al. (2005)*. Children were asked to close their incisor teeth edge to edge. Cheeks and lips were retracted so that all anterior teeth and some parts of upper and lower gums were visible. The camera was adjusted to 15° above the perpendicular to the central incisors' plane to minimize specula reflection and burn outs (*Ellwood, Cortea & O'Mullane, 1996*). Immediately after taking each photograph, it was assessed to confirm its quality, and was repeated if necessary.

During the fluorosis assessment phase, first, the photographs were randomly ordered to prevent bias induced by the assessors' foreknowledge of the fluoride in the area. Photographs were then assessed by eight calibrated dentists who were blind to the clinical condition and town of residence of the subjects. Four calibrated dentists assessed the photographs based on Dean's Index, and another four used the TF Index (*Dean, 1942*; *Fejerskov, 1988*). All dentists observed the photographs on one computer with identical settings. The diagnosis of fluorosis was confirmed only if three out of four dentists of each group agreed.

The objective of the next stage of study was to obtain data on early childhood adverse events. The parents of all children participating in the study were invited to school for interview. Name and date of birth of each child were double-checked with their parents. Parents were interviewed using the ECEL method (*Golkari, 2009*). Information regarding gestational age (preterm, term, delayed), birth weight, number of births, type of delivery (natural, caesarean or facilitated delivery), trauma to baby during birth, and newborn vitality score was obtained. Questions were asked relating to personal life line, residential status, occupation of parents/guardians, and child activity line which could help parents to remember adverse health-related events. Afterwards, any illnesses the child had suffered were recorded. For each illness the parent was asked about the name or description of illness, age at which the illness started, duration, perception of severity (mild, moderate, or severe), if went to doctor, medication if used, and hospitalization. Information regarding hospitalizations (age, duration, reason, type of anesthesia if used, and name of hospital), and falls and accidents (age, cause, trauma to face or teeth, hospitalization, and breathing status right after accident as an indicator of severity well known by parents) were also obtained.

The timing of childhood adverse health-related conditions was matched with the timing of formation and calcification of each part of permanent incisors (*Golkari, 2009*). If no adverse life condition could be matched to the position of a defect diagnosed as fluorosis, the case was considered as genuine fluorosis. However, if a health-related adverse condition could be matched to an enamel defect diagnosed as fluorosis, the case was considered as

**Table 1** The comparison of fluorosis scores according to Dean's Index by four calibrated dentists.

|  | Test | Examiner 2 | Examiner 3 | Examiner 4 |
|---|---|---|---|---|
| Examiner 1 | Kappa value | 0.16 | 0.16 | 0.07 |
|  | McNemar *p*-value | <0.001 | <0.001 | <0.001 |
|  | Correlation coefficient | 0.528 | 0.454 | 0.458 |
| Examiner 2 | Kappa value |  | 0.34 | 0.37 |
|  | McNemar *p*-value |  | 0.125 | <0.001 |
|  | Correlation coefficient |  | 0.584 | 0.575 |
| Examiner 3 | Kappa value |  |  | 0.29 |
|  | McNemar *p*-value |  |  | 0.003 |
|  | Correlation coefficient |  |  | 0.614 |

fluorosis-resembling defect. Using this method, diagnosed fluorosis defects were divided into genuine fluorosis and fluorosis resembling defects.

SPSS software (version 22) was used for data analysis. Agreements between examiners who assessed fluorosis using Dean's Index and TF Index were tested using Kappa-coefficient. McNemar, and Pearson correlation tests were also used to compare the results reported by each pair of dentists. The prevalence of fluorosis in the three selected areas was compared, before and after adjustments for sex, by chi-square test and logistic regression. Childhood adverse health-related conditions were matched to enamel defects diagnosed as fluorosis one by one.

## RESULTS

The number of 9-year-old children included in the study was 376; 171 (46%) girls and 196 (53 %) boys. The number of included children from Gerash (high F), Shiraz (optimal F) and Sepidan (low F) were 88, 189, and 90 respectively.

Using both Dean and TF indices, three out of four dentists diagnosed that 31.3% (115) of children had fluorosis. The percentage of fluorosis cases in areas with high, optimal, and low range of fluoride in water was 58.0%, 29.1%, and 10.0% respectively ($p < 0.001$). Logistic regression showed that there was a positive relationship between fluorosis and fluoride in the drinking water ($p < 0.001$). There was no relationship between fluorosis and children's sex ($p = 0.228$).

There were significant differences among dentists who scored photographs using Dean's and TF indices. Among the four dentists who assessed photographs according to Dean's Index, the difference in the number of cases diagnosed as fluorosis was statistically different between each two dentists ($p < 0.001$). There was only a slight (kappa was between 0 and 0.2) or fair (kappa was between 0.2 and 0.4) agreement between them (Table 1). Similar results of agreement were observed among the four dentists who scored children using the TF Index. Although there was not a high agreement among dentists, a positive correlation was observed ($p < 0.001$) (Table 2).

Adverse past childhood health-related events were found in 311 (84.7%) of children using the ECEL method. Fluorosis defects were found in 115 (31.3%) of children. The

**Table 2** The comparison of fluorosis scores according to TF Index by four calibrated dentists.

|  | Test | Examiner 2 | Examiner 3 | Examiner 4 |
|---|---|---|---|---|
| Examiner 1 | Kappa value | 0.06 | 0.21 | 0.34 |
|  | McNemar *p*-value | <0.001 | <0.001 | <0.001 |
|  | Correlation coefficient | 0.381 | 0.465 | 0.498 |
| Examiner 2 | Kappa value |  | 0.27 | 0.18 |
|  | McNemar *p*-value |  | <0.001 | <0.001 |
|  | Correlation coefficient |  | 0.472 | 0.503 |
| Examiner 3 | Kappa value |  |  | 0.36 |
|  | McNemar *p*-value |  |  | <0.001 |
|  | Correlation coefficient |  |  | 0.539 |

timing of adverse health-related events could be matched with the position of what was diagnosed as fluorosis in 31 (8.4%) of children. These were regarded as fluorosis resembling defects. Therefore, it was concluded that 26.9% of what was first diagnosed as fluorosis was in fact fluorosis resembling defects.

The overall percentage of genuine fluorosis was 22.9%: 47.7%, 20.6%, and 3.3% in areas with high, optimal, and low fluoride areas respectively ($p < 0.001$). The percentage of fluorosis resembling defects in areas with high, optimal, and low range of fluoride in water was 10.2%, 8.5%, 6.7% respectively. The difference in percentage of fluorosis resembling defects among the three areas was not statistically significant (Table 3).

## DISCUSSION

Fluoride is one of the most successful measures for prevention of dental caries in public health (*Petersen & Lennon, 2004*). However, there has always been controversy about using fluoride because of fluorosis (*Sapolsky, 1968*; *Null & Feldman, 2003*; *Ananian, Solomowitz & Dowrich, 2005*). Reports of a high prevalence of fluorosis in communities have led to objections to fluoride. Therefore, there is a need for a precise way to diagnose dental fluorosis. Many local and systemic risk factors cause DDEs. Some non-fluoride related DDEs are similar to enamel fluorosis and should be differentiated from genuine fluorosis. No standard method has been established to differentiate them from one another. The main objective of this study was therefore, to try a method to distinguish fluorosis from other kinds of DDEs that look like fluorosis but are caused by adverse health-related events, not by excess fluoride intake.

A systematic review reported that in Iran with an average concentration of fluoride in water of $0.43 \pm 0.17$ ppm, the prevalence of fluorosis was 61% (*Azami-Aghdash et al., 2013*). Based on the level of fluoride, the reported prevalence of fluorosis was high and questionable. The inconsistency of the prevalence of fluorosis with the level of fluoride in water reported in that study and in many other different parts of the world (*Lo & Bagramian, 1996*; *Sudhir et al., 2009*; *Arif et al., 2013*) suggests that dental fluorosis was misdiagnosed.

The overall prevalence of diagnosed dental fluorosis in the current study was 31.3%. However, by using the ECEL method and considering health-related events, the prevalence

**Table 3** Comparison of all defects diagnosed as fluorosis, genuine fluorosis, and fluorosis-resembling defects among the three areas.

|  |  | All children diagnosed with fluorosis[a] | Children with genuine fluorosis | Children with fluorosis-resembling defects |
|---|---|---|---|---|
| Level of fluoride in area | High (N = 88) | 51 (58.0%) | 42 (47.7%) | 9 (10.2%) |
|  | Optimal (N = 189) | 55 (29.1%) | 39 (20.6%) | 16 (8.5%) |
|  | Low (N = 90) | 9 (10.0%) | 3 (3.3%) | 6 (6.7%) |
|  | OR optimal/high (95% CI) | 0.303[*] (0.179–0.514) | 0.292[*] (0.168–0.506) | 0.797 (0.337–1.886) |
|  | OR low/high (95% CI) | 0.080[*] (0.036–0.180) | 0.037[*] (0.011–0.127) | 0.625 (0.231–1.837) |
| Sex | Boys (N = 196) | 67 (34.2%) | 52 (26.5%) | 15 (7.7%) |
|  | Girls (N = 171) | 48 (28.1%) | 32 (18.7%) | 16 (9.4%) |
|  | OR girls/boys (95% CI) | 0.744 (0.462–1.201) | 0.619 (0.362–1.056) | 1.251 (0.597–2.620) |

**Notes.**
[a] All children that were diagnosed as having fluorosis by three out of four calibrated dentists using both Dean and TF Indices.
*Significant at 0.001 level.

of genuine dental fluorosis was 22.6% as there were 8.4% with fluorosis-resembling defects. That illustrates that fluorosis could be ruled out as the main cause of about 27% of what was incorrectly diagnosed as fluorosis.

The current study also showed significant differences among the dentists who scored photographs to diagnose fluorosis according to Dean's and TF Indices ($p < 0.001$). This finding indicates that both Dean and TF Indices are too subjective and therefore not precise ways to diagnose dental fluorosis.

These two Indices could lead to misdiagnosis of fluorosis-resembling defects as genuine fluorosis. *Tavener, Davies & Ellwood (2007)* also concluded that interpretation of criteria could be different among examiners and stressed the necessity for standard methods to measure dental fluorosis. Some studies have shown good to excellent agreement among examiners (*Kumar et al., 2000*). However, comparing the methods, the lack of bias was an advantage of the current study, as photographs were assessed instead of clinical examinations.

This study indicates that by considering adverse health-related events, it is possible to distinguish genuine fluorosis from fluorosis-resembling defects. When the timing of an adverse condition matches the timing of development of the part of enamel defect, the adverse event could be the cause of the defect, not fluoride. If no adverse condition could be matched to a defect, excess fluoride could, with caution, be considered as the cause. In case of generalized defects, fluoride, genetic, or severe underlying systematic disease, either individually or as a combination could be the cause.

One limitation of this study is that even if an adverse health-related event could be exactly matched to a fluoride-resembling defect in terms of time and place, it could not be definitely considered that the adverse event was the cause of the defect and fluoride was not the cause. On the other hand, if an adverse condition could not be matched to a supposedly fluorosis defect, one could not be sure that fluoride was the cause. In fact, the definite diagnosis of fluorosis is not possible by this method and the only exact way to diagnose dental fluorosis would be by microscopic or chemical analysis. However,

although the ECEL method cannot prove that fluoride is or is not the cause of defect, it can help in differentiate between genuine fluorosis and fluorosis-resembling defects. In the case of this study, past childhood data obtained by the ECEL method helped to find fluorosis-resembling defects in 10.2%, 8.5%, and 6.7% of subjects from high, optimal, and low fluoride areas, respectively.

It was decided from the beginning of the study that children with large restorations on their permanent anterior teeth should be excluded as it was not possible to assess the previous existence of defects on them. This was while the authors were aware of the bias this could make. The large restorations could be due to fluorosis, non-fluoride related DDEs, dental caries, trauma or other reasons. One could rule out dental caries and trauma easily by questioning the parents. However, there was no means of distinguishing between fluorosis and any other types of defects. Fortunately, in this study, only one child was excluded based on having large restorations. This one subject could not affect the results of this study. However, this should be addressed in future studies.

The diagnosis of fluorosis is more complicated than acknowledged. On the other hand, there has always been considerable controversy regarding the use of fluoride for dental caries prevention. One of the main issues, which opponents of fluoridation raise, is fluorosis. Therefore, finding a reliable way for more accurate diagnosis of genuine fluorosis is vital. Existing fluorosis indices could lead to misjudgment about using fluoride. However, the ECEL method to record health–related life events is a promising method to help differentiate between genuine fluorosis and fluorosis-resembling defects.

## CONCLUSION

Fluorosis indices, if used alone, could result in misdiagnosis of dental fluorosis and misguide health policymakers in their decision about public health measure related to use of fluoride. Information about adverse health-related conditions linked to DDEs at specific positions on teeth could help to differentiate between genuine fluorosis and fluorosis-resembling defects.

## ACKNOWLEDGEMENTS

This manuscript is based on ASa PhD Thesis which was registered in Shiraz University of Medical Sciences (# PhD 122) and was conducted under supervision of AG and advisory of ASh. The authors would like to thank Dr. Mehrdad Vossoughi for his statistical advice.

### Funding

Research supported by the Vice-Chancellery for Research of Shiraz University of Medical Science (Grant # 93-7770). The funders had no role in study design, data collection and analysis, decision to publish, or preparation of the manuscript.

## Grant Disclosures

The following grant information was disclosed by the authors:
Vice-Chancellery for Research of Shiraz University of Medical Science: 93-7770.

## Competing Interests

The authors declare there are no competing interests.

## Author Contributions

- Aira Sabokseir conceived and designed the experiments, performed the experiments, analyzed the data, wrote the paper, prepared figures and/or tables.
- Ali Golkari conceived and designed the experiments, analyzed the data, wrote the paper.
- Aubrey Sheiham conceived and designed the experiments, reviewed drafts of the paper.

## Human Ethics

The following information was supplied relating to ethical approvals (i.e., approving body and any reference numbers):

Shiraz University of Medical Sciences, Postgraduate Faculty, Approval Number: 93/42937.

## Data Availability

The raw data was provided as Data S1.

## Supplemental Information

Supplemental information for this article can be found online at http://dx.doi.org/10.7717/peerj.1745#supplemental-information.

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
