# Peer review of "Distinguishing between enamel fluorosis and other enamel defects in permanent teeth of children"

_PeerJ, doi:10.7717/peerj.1745_

## Round 0.1 · original submission · Minor Revisions

Please address the comments from the reviewers.

Reviewer 1 ·

Basic reporting

Research question is good
Language needs improvement in introduction.
Literature is relevant and well referenced

Experimental design

Primary research is within the scope of the journal
what is the age group of the subjects (whether it is 0 to 9 year or only 9 year old).
There will be Memory/Recall bias with the ECEL Method

Validity of the findings

Findings are valid as per the method
There are no ANOVA tables since authors have mentioned about the ANOVA
Tables are not labelled properly and no foot notes on each table
conclusion is well stated

Additional comments

No comments

Reviewer 2 ·

Basic reporting

- include mention of the varying fluoride levels in the abstract results since you mentioned them in the abstract methods regarding the selection process
- in the INTRO (lines 46-47): can you clarify/expand upon the following sentence (perhaps by tracking these three themes through the rest of the paragraph more clearly/with the same terms)?: "The differences between some of the diagnostic categories are uncertain, vague, or insensitive."
- can you finish this sentence?: "Still the differentiation between some categories of TFI is not sufficiently precise....... (for what?)"
- although this journal does not emphasize impact or novelty, it does emphasize relevance and meaning: I think it would strengthen your manuscript to articulate throughout the importance and implications of your research question, to explain why the following statements (to list a few) matter: "need for accurate data of dental fluorosis" (line 83), "However, there has always been controversy about using fluoride because of fluorosis…" (lines 209-210), " Therefore, there is a need for a precise way to diagnose dental fluorosis" (line 212).
- please describe the various appearances of DDEs and show how fluorosis can be easily confused with non-fluoride related DDEs

Experimental design

- please briefly describe HOW fluorosis is described/identified by each index so the reader can have some insight into the subjectivity inherent in using these measures
- should state HOW you confirmed the participants had lived in their respective area when they were less than ~6 years of age
- should state how you know the water source (and thus fluoride exposure) is the same for all participants in each region
- should explain why you chose communities with fluoride up to 1.22 for the "optimal fluoride " category if 1 ppm is the upper limit of "optimal" (as stated in the same paragraph, line 104 v. line 107)
- exclusion of children - if the reasons for exclusion are related to DDEs (eg, large restorations could result from severe defects), could bias results; acknowledge this in the Discussion section and discuss the expected direction of such bias
- discuss "recall bias" as a limitation of the ECEL method, especially for recording illnesses (typical illnesses – opposed to major accidents, etc. -- may be particularly difficult to remember 4-9 years after the fact; discuss the expected direction of this bias (ie, what if an illness was not recalled/forgotten?, what if an illness was recalled at the wrong point in time?)

Validity of the findings

- be cautious and perhaps more conservative with the wording of and assumptions around fluorosis and DDEs throughout the manuscript; for example, “That suggests that what was diagnosed as fluorosis were DDEs which may resemble fluorosis." (lines 30-31) may be better worded as: "That suggests that what was diagnosed as fluorosis MAY HAVE BEEN NON-FLUORIDE/OTHER DDEs THAT resemble fluorosis/FOLLOW A SIMILAR PATTERN TO FLUOROSIS."
- be careful not to suggest DDEs exclude fluorosis (fluoride is one cause of DDEs) and "fluorosis" is just a subjective categorization of defects based on assumptions, as your paper nicely demonstrates. Be sure to make this clear and consistent throughout, especially for non-dental readers.
- I think the 4th paragraph of the RESULTS (lines 195-205) needs to be revised. It was very confusing to me and these are the main results! More specifically, is 8.4 really the % of both 1) all children who had fluorosis-resembling defects (line 195) and 2) of children diagnosed with fluorosis, those who matched with health-related events (line 197; 31/115 = 2.7% ???). If I have misunderstood, I think this needs clarifying. If this is a strange coincidence, please acknowledge this so the reader will not be confused.
- the 4th paragraph of the RESULTS (lines 195-205) should consistently present whole numbers (number of participants) followed by percentages in parentheses; please avoid only presenting results as percentages
- please clarify the following sentence (lines 201-202): "The fluorosis resembling defects (8.4% of all children) included 26.9% of what was diagnosed as fluorosis by Dean’s and TF Indices."
- discuss the expected direction of bias for each scenario of the described limitations (lines 251-259)
- I think discussion/mention of the 8.5% and 6.7% final fluorosis prevalence estimates in the optimal and low settings warrants notice/mention (perhaps tie this into the existing discussion about limitations on lines 251-259)
- it would strengthen the article to directly state the potential public health implications of the following phrase (line 261): "misjudgment about using fluoride."
- similarly, WHY is the stated Conclusion important (again, directly state the public health implications?)?

Additional comments

I appreciate your creative approach to a critical question that has been swept under the carpet but is worthy of critical evaluation. Please be sure to bolster your manuscript with clearer articulation of the context of your research question and the potential public health implications of your findings!

- after line 233 or 240 I was expecting and looking forward to suggestions for improved (more objective) measurement of fluorosis and/or DDEs in general (other scale, such as DDE index?) after such a critical evaluation of the existing scales/indices/approach. This question will be of particular interest to those who do not have ECEL data. Or, can fluorosis not really be measured without laboratory/radiographic methods?
- Please clarify what the “difference” is of (lines 214-215): "The difference has not yet been evaluated"
- citation needed for the following (line 218): "A systematic review reported that the prevalence of fluorosis in Iran was 61%"
- please clarify the middle of the sentence on line 247 ("such defects should not be seen on the teeth of children who experienced the same adverse health– related conditions and....")

·

Basic reporting

Clear, unambiguous, professional English language used throughout.
Intro & background to show context. Literature well referenced & relevant.
Structure conforms to PeerJ standard, discipline norm, or improved for clarity.
Figures are relevant, high quality, well labeled & described.
Raw data supplied.

Experimental design

Negative/ inconclusive results accepted.
Meaningful replication encouraged where rationale & benefit to literature is clearly stated.
Data is robust, statistically sound, & controlled.

Validity of the findings

Research question well defined, relevant & meaningful. It is stated how research fills an identified knowledge gap.
Rigorous investigation performed to a high technical & ethical standard.
Methods described with sufficient detail & information to replicate.
Conclusion well stated, linked to original research question & limited to supporting results.

Additional comments

The report is concise and important in its field. Data is robust and study has a rigorous method. It is acceptable for publication with minor English revision.
Comments: Please write complete words such as (TF) for the first time.
Was there any difference between fluorosis resembling defects among three area with different water fluoride level?

---

## Round 0.2 · accepted · Accept

The reviewer comments appear to be adequately addressed. Congratulations!